# Global assessment of storm disaster-prone areas

**Nazzareno Diodato[1], Pasquale Borrelli[2,3], Panos Panagos[4]\*, Gianni Bellocchi[1,5]**

**1** Met European Research Observatory—International Affiliates Program of the University Corporation for Atmospheric Research, Benevento, Italy, **2** Department of Science, Roma Tre University, Rome, Italy, **3** Department of Biological Environment, Kangwon National University, Chuncheon, Republic of Korea, **4** European Commission, Joint Research Centre (JRC), Ispra, Italy, **5** Université Clermont Auvergne, INRAE, VetAgro Sup, UREP, Clermont-Ferrand, France

\* Panos.PANAGOS@ec.europa.eu

## Abstract

### Background

Advances in climate change research contribute to improved forecasts of hydrological extremes with potentially severe impacts on human societies and natural landscapes. Rainfall erosivity density (RED), i.e. rainfall erosivity (MJ mm hm$^{-2}$ h$^{-1}$ yr$^{-1}$) per rainfall unit (mm), is a measure of rainstorm aggressiveness and a proxy indicator of damaging hydrological events.

### Methods and findings

Here, using downscaled RED data from 3,625 raingauges worldwide and log-normal ordinary kriging with probability mapping, we identify damaging hydrological hazard-prone areas that exceed warning and alert thresholds (1.5 and 3.0 MJ hm$^{-2}$ h$^{-1}$, respectively). Applying exceedance probabilities in a geographical information system shows that, under current climate conditions, hazard-prone areas exceeding a 50% probability cover ~31% and ~19% of the world's land at warning and alert states, respectively.

### Conclusion

RED is identified as a key driver behind the spatial growth of environmental disruption worldwide (with tropical Latin America, South Africa, India and the Indian Archipelago most affected).

## Introduction

Although there is a growing need to assess ecosystem responses to climate change-induced disaster risk reduction, there is a lack of research on sensitive areas and on coastal zones, drylands and watersheds, particularly in Global South low-income countries [1, 2]. In order to advance global climate change studies, climate impact indicators are needed to help develop

**Data Availability Statement:** The data can be requested using the request form through the following link: https://esdac.jrc.ec.europa.eu/content/global-rainfall-erosivity.

**Funding:** The authors received no specific funding for this work.

**Competing interests:** The authors have declared that no competing interests exist.

guidelines for landscape conservation planning [3, 4] and support decision-making based on hydrological damage data [5, 6]. Rainfall erosivity density (RED in MJ hm$^{-2}$ h$^{-1}$, equivalent to the most common MJ ha$^{-1}$ h$^{-1}$), i.e. rainfall erosivity per unit of rainfall, is an important climatic indicator of floods and soil erosion [7, 8]. RED effects are apparent in catastrophic weather events, due to the intensification of daily storms in the northern Hemisphere (**Fig 1A**) and erosive precipitation in other parts of the world (**Fig 1B**). The effects of more intense extreme rainfall worldwide [9–12] include intensified sub-daily precipitations [13, 14], the local occurrence of flash floods [15] and the incidence of the erosive force of rainfall [16–19].

Damaging hydrological events are extreme phenomena, the source of multiple hazardous events with potentially serious impacts on human societies [21]. Among these, floods and events causing flooding are identified as hydrological disasters, which also include meteorological disasters like thunderstorms. Like other extreme phenomena, hydrological disasters typically leave behind socio-economic damage, the severity of which depends on the resilience of the affected population and the available infrastructures [22]. Potentially severe natural events are typically not classified as natural disasters if they occur in areas without a vulnerable population, e.g. deserts [23]. Information on the spatial distribution of RED can help delineate areas prone to multiple damaging hydrological events [24, 25] and, in turn, can transfer important prescriptions for disaster response planning. However, geoinformation and hazard mapping have been and remain critical in modern visual communication science practice [26], particularly in landscape decision-making, where the uncertainty component must be included in the mapping [27, 28]. In particular, it is difficult to represent disaster-prone areas by RED, as different parts of terrestrial ecosystems respond differently to uneven, often nonlinear and universal forcing agents, with threshold-like features [29, 30]. Landscapes are, in fact, highly responsive and non-linear systems to both external dynamics, such as climatic and non-climatic factors, with a combination of gradual changes coupled with infrequent high-magnitude events has led to dramatic landscape responses throughout Earth's history [31]. Then, the effects of these historical landscape responses to climate extremes and threshold processes are key parameters affecting the geomorphological impacts of extreme hydrological events [32, 33].

Threshold mapping, and the processes of developing probability mapping, is a challenge in geographic information science and spatial downscaling [34]. Scaling and integrating the relative uncertainty of these thresholds, for which storms drive surface flows, including

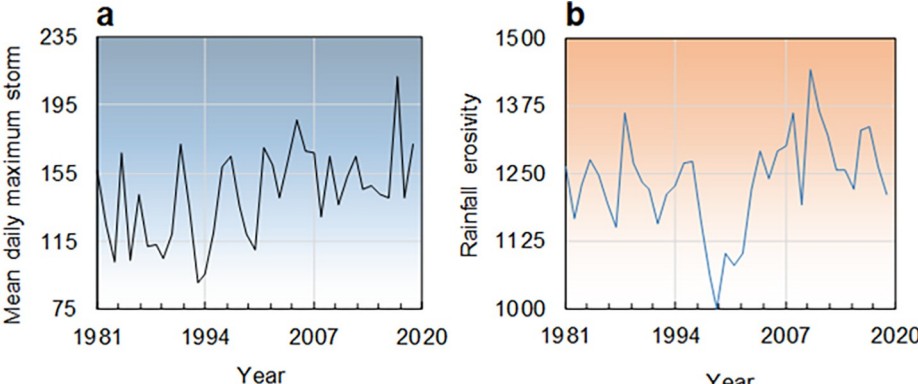

**Fig 1. Emerging catastrophic weather events over the last four decades.** a) Annual mean daily maximum storm depth (mm d$^{-1}$) in the Northern Hemisphere (mean data of areal maxima from NCEP/NCAR Reanalysis [20]), and b) Global evolution of the erosive force of rainfall (rainfall erosivity) in MJ mm hm$^{-2}$ h$^{-1}$ yr$^{-1}$ (arranged from Bezak et al. [19]). Graph a) covers the period 1981–2018, and graph b) covers the period 1981–2020.

geomorphic effectiveness and erosional soil degradation, affect the spatial pattern of damaging hydrological events, and ultimately improve our understanding of ecological responses to climate extremes and thresholds [35].

The apparent naturalness of a landscape measures the degree to which it is free from storm damage (Fig 2A). Under erosive climatic conditions, the strength, intensity and frequency of a given rainfall influence hydrological processes, even though they may still maintain the shape of the land and the equilibrium of the environmental system in a resilient landscape (Fig 2B). On the contrary, a change in the hydrological regime, especially when the thresholds of an acceptable level of hydrological disturbance are exceeded, can have harmful consequences for the landscape (Fig 2C).

The prerequisite for RED modelling is that the environmental system adapts to changes in the natural hydrological regime. Usually, the prediction of precipitation and its extremes is performed using physical models, mainly due to the high spatial variability and nonlinearity of the problem [36]. The limitations of physical models (which are mainly computational at the global level) are encountered with very extreme rainfall, when the prediction has to be performed at time-scales less than 30 minutes over many years and for a spatial resolution of a few kilometres. Recently improved convection models can predict extreme rainfall, but simulations are not currently within reach due to their computational cost and degree of uncertainty [37]. Retrieving extreme rainfall from satellite data would be an option to improve the estimation of rainfall erosivity [38], but satellite data require corrections and need further assessment, especially on a global scale [39]. With the advancement of global atmospheric reanalysis data, numerical weather prediction techniques are becoming an encouraging mean of estimating rainfall erosivity [40]. In particular, the production of global RED maps is a challenge due to two conflicting conditions, namely that the analysis requires precipitation data with both high temporal resolution and global coverage. With a large volume of data and uneven distribution of stations, geostatistical methods provide reasonable estimates of what the variable of interest would be at intermediate locations. Geostatistics offers different approaches to deal with this issue and provides attracting results when experimentally determined rainfall erosivity data are available at both regional [41, 42], continental [43, 44], or even global scales [5]. However, geostatistical estimation of RED [17, 45, 46], and its spatial patterns above given thresholds [47], have generally received limited attention [48].

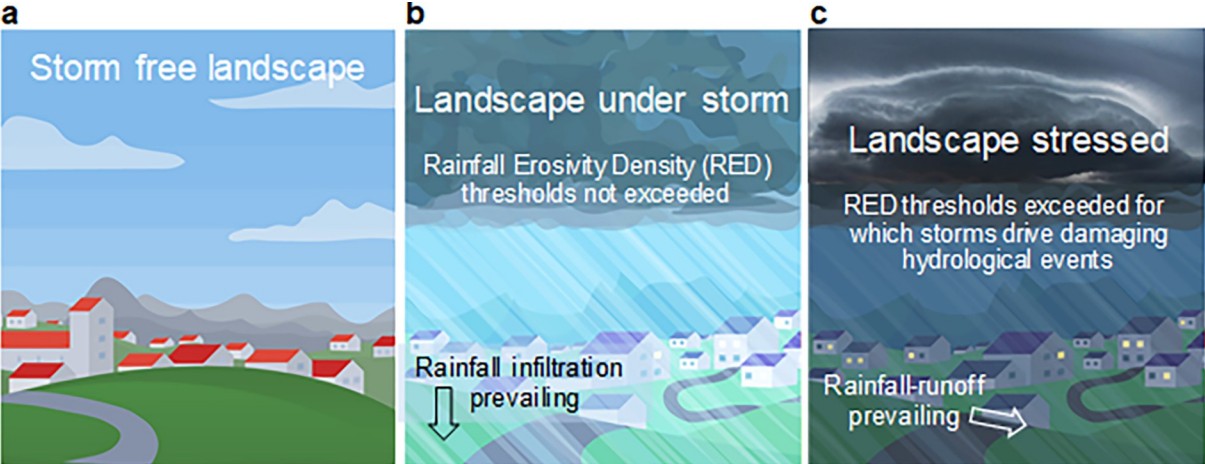

**Fig 2.** Landscape under different weather regimes with changing RED thresholds for a given level of hydrological disturbance in: a) Storm-free landscape, b) Landscape under storms not exceeding thresholds, and c) Stressed landscape where thresholds are exceeded resulting in damaging hydrological events (image arranged from MeteoBlue, https://static.meteoblue.com/assets/images/crosslinks/yearcomparison.svg).

The value of geostatistical probability mapping for geographic information systems (GIS) lies in providing reliable interpolation methods in error assessment and scaling, which can be used in environmental modelling and decision-making [49]. The spatial uncertainty associated with RED hazard over a range of scales is, in any case, an open issue, coupled with uncertainties in downscaling methods and a lack of primary information data in many areas. The uncertainty of RED thresholds actually poses challenges for geospatial assessment, as the worst storms can fall at locations not well-covered by weather recording stations [50]. Global quantification of storms at fine time scales also remains challenging, as hourly or sub-hourly rainfall data of sufficient length are poor, especially for critical and vulnerable regions such as the tropics [51].

Here, we present for the first time the use of the Global Rainfall Erosivity Database (GLoREDa) to estimate RED worldwide [5]. For this study, this database was updated as GLoREDa-V2 and contains erosivity values estimated as RED from 3,625 stations in 63 countries with time resolutions from 1 to 60 minutes. The results obtained are based on climate data and a probabilistic approach to proceed under a *soft* geovisualisation in order to mitigate the uncertainty involved in downscaling and geocomputational tracking. For this purpose, we used a parametric kriging technique, which provides great flexibility for modelling environmental data [52]. In particular, GIS and log-normal ordinary kriging coupled with output probability mapping (LNOKpm) were used to continuously delineate the spatial uncertainty of RED thresholds and predict areas prone to damaging hydrological events on a global scale.

## Methods

### Rainfall erosivity data

We refer to annual rainfall erosivity data from the Global Rainfall Erosivity Database (GloREDa, here updated to GloREDa-V2), which covers 3,625 precipitation stations from 63 countries with temporal resolutions of 1 to 60 minutes. It is the result of an extensive collection of high temporal resolution rainfall data from as many countries as possible in order to have a representative sample across climatic and geographical gradients (**Fig 3**).

The number of stations varies greatly from continent to continent, with no station-data available above 70° North and below 47° South. However, the latter is not an issue. In fact, the heaviest rains have a low probability of occurring over the northern limit, where rainfall erosivity values are close to zero, while large areas below the southern limit are open sea water or ice-covered (Antarctic ice cap). Precipitation time-series range from a minimum of 5 years to maximum of 52 years (on average, 16.8 years).

GloREDa contains the best available global datasets of annual rainfall erosivity (RE, MJ mm hm$^{-2}$ h$^{-1}$ yr$^{-1}$), in the form of (R)USLE-R factor [53], calculated on a monthly ($j = 1, \ldots, 12$) basis (RE$_m$, MJ mm hm$^{-2}$ h$^{-1}$ month$^{-1}$), from which we obtained rainfall erosivity density (RED, MJ hm$^{-2}$ h$^{-1}$) as the ratio between rainfall erosivity and precipitation amount [54]:

$$\text{RE}_j = \frac{1}{n} \cdot \sum_{i=1}^{n} \sum_{k=1}^{m_j} \left\{ I_{30} \cdot \sum_{r=1}^{m} \left[ 0.29 \cdot \left( 1 - 0.72 \cdot e^{-0.05 \cdot i_r} \right) \right] \cdot v_r \right\} \cdot k \tag{1}$$

$$\text{RE}_y = \sum_{j=1}^{j=12} \text{RE}_j \tag{2}$$

$$\text{RED} = \frac{\text{RE}}{\frac{1}{n} \cdot \sum_{i=1}^{n} P_y} \tag{3}$$

where $n$ is the number of years recorded; $m_j$ is the number of erosive events during a given

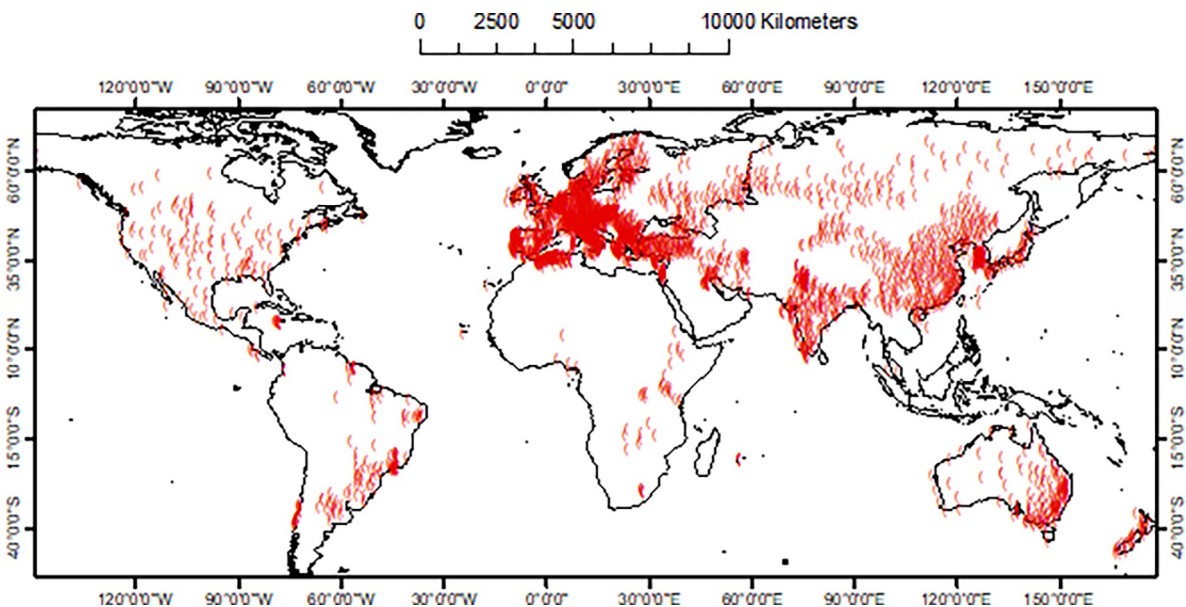

**Fig 3. Geographical extent of the global network of stations (red circles) where rainfall erosivity density (RED) data were available.**

month $j$; $k$ is the index of a $k^{th}$ single event; $v_r$ is the volume of rainfall (mm) during the $r^{th}$ period of a storm, which splits into $m$ parts; $I_{30}$ is the maximum 30-minute rainfall intensity (mm h$^{-1}$); $i_r$ is the rainfall intensity during the time interval (mm h$^{-1}$); $P_y$ is the amount of rain (mm) during a given year $y$. The numerator in Eq (3) represents the long-term mean rainfall erosivity (MJ mm hm$^{-2}$ h$^{-1}$ yr$^{-1}$), while the denominator is the relative mean amount of precipitation (mm) over the same number of years.

## Warning and alert thresholds

We use concepts of a "warning state" and an "alert state" [55] for erosivity density, as regions with high RED are at risk of flooding and water scarcity owing to their infrequent but very intense and erosive rainstorm [19, 56]. As such, RED reflects not only the component of climate forcing reproduced in the aggressiveness of the storm (rainfall erosivity), but also the damaging hazard and its associated hydrological risk [18]. In particular, RED values > 3.0 MJ ha$^{-1}$ h$^{-1}$ indicate an increased risk of erosive rainstorms, soil erosion and flooding [57]. This critical threshold value was popularised in storm geomorphology by Dabney et al. [58] as the runoff increases when RED exceeds 3 MJ hm$^{-2}$ h$^{-1}$, thus leading to an increasing erosive hazard as storm erosivity represents a large proportion of the rainfall amount in an intense event. Mostly set regionally on a monthly basis [44], this hydrological threshold (alert) could help detecting areas prone to erosion- and overland flow [59, 60]. Abstractions such as thresholds of change and strength are all essential as landscapes may be able to counter or assimilate pulses of change as a form of sensitivity or stability [61]. The use of this abstraction for the way geomorphological systems react to climate variability and change is still an important topic since its conception by Allison and Thomas [62] and Phillips [63]. Given the global scale and annual resolution of this study, a second, smaller RED threshold value (warning) was also used to better capture interactions between changing hydrologic variability and ecological responses. For instance, major tropical forest regions like Amazon and African regions are less vulnerable to hydrological hazards than Southeast Asia [64]. When RED values are > 1 MJ ha$^{-1}$ h$^{-1}$, only a certain amount of precipitation can cause relatively high erosivity [57]. Therefore,

we adopted two threshold values as follows: RED exceeds the threshold of 1.5 MJ hm$^{-2}$ h$^{-1}$ in the warning state and exceeds 3.0 MJ hm$^{-2}$ h$^{-1}$ in the alert state. The two values correspond, respectively, to the median and the 3$^{rd}$ quartile of the distribution of the RED data (Fig 4A). These values also mark the range of critical RED values identified by Diodato et al. [8] with a 50-year return period in north-western Italy. The advantage of using RED threshold values compared to other rainfall aggressiveness indicators are [54]: a) greater propensity to classify geomorphological hazards; b) greater stability, obtained with a shorter and heterogeneous period of recording; and c) easier mapping over large areas due to its independence on altitude up to 3000 m a.s.l. [65].

## Log-normal ordinary kriging probability map

Kriging is a generic name covering a range of spatial least-square prediction methods [66]. Most kriging algorithms with references to hydrogeological applications were reviewed in Kitanidis [67]. For some kriging models, only one estimate per cell is required; for others, as in decision-making, it is necessary to know the local uncertainty associated with the estimates [68, 69]. The ordinary lognormal kriging algorithm has the potential to improve maps because such techniques can be combined with data transformation and detrending options, and can take different forms, such as maps of probability outputs, quantiles and standard errors of prediction, when the normality of the distribution is verified [52]. In contrast, indicator kriging cannot perform data-transform and detrending, when *soft* information is required within the probability mapping.

If the prediction in the unknown locations is normally distributed, then the mean and median of the predictions will be positioned centrally in the probability density distribution of each location.

The area under the distribution curve to the right of the threshold line predicts the probability that the value is greater than a threshold value. The distribution of predictions changes for each location as the mean and standard error of the predictions change. Thus, by keeping the threshold value constant, a probability map is produced for the whole area. Since the RED data show a skewed distribution, we used a lognormal ordinary kriging in the form of a probability output map (LNOKpm). The assumption of normality of the distribution was checked after the data were log-transformed. A straightforward approach is to classify as hazardous all locations where the probability of exceeding a critical threshold value, $z_k$, is greater than a critical RED value (1.5 MJ hm$^{-2}$ h$^{-1}$ for the warning state and 3.0 MJ hm$^{-2}$ h$^{-1}$ for the alert state). The ordinary kriging model assumes that the data are a realisation of an auto-correlated process plus an indipendent random error. For a complete analytical procedure of LNOKpm, refer to Krivoruchko et al. [70].

# Results and discussion

## Exploratory data analysis and transformation

Exploratory data analysis is important to inspect and explore data statistically before deciding whether and how to transform them for analysis and to illustrate what can be achieved by transforming data (e.g. into logarithms) of single variates and calculating principal component analysis of multivariate data [71]. The first step in the spatial analysis is to check the raw data for drifts and outliers [72], and finally for normal distribution. Outliers and spatially-skewed data can be detected by the frequency distribution and the third standardised moment or skewness (*g*), whose value is 0 for a normal distribution and any other symmetric distribution with finite third moment.

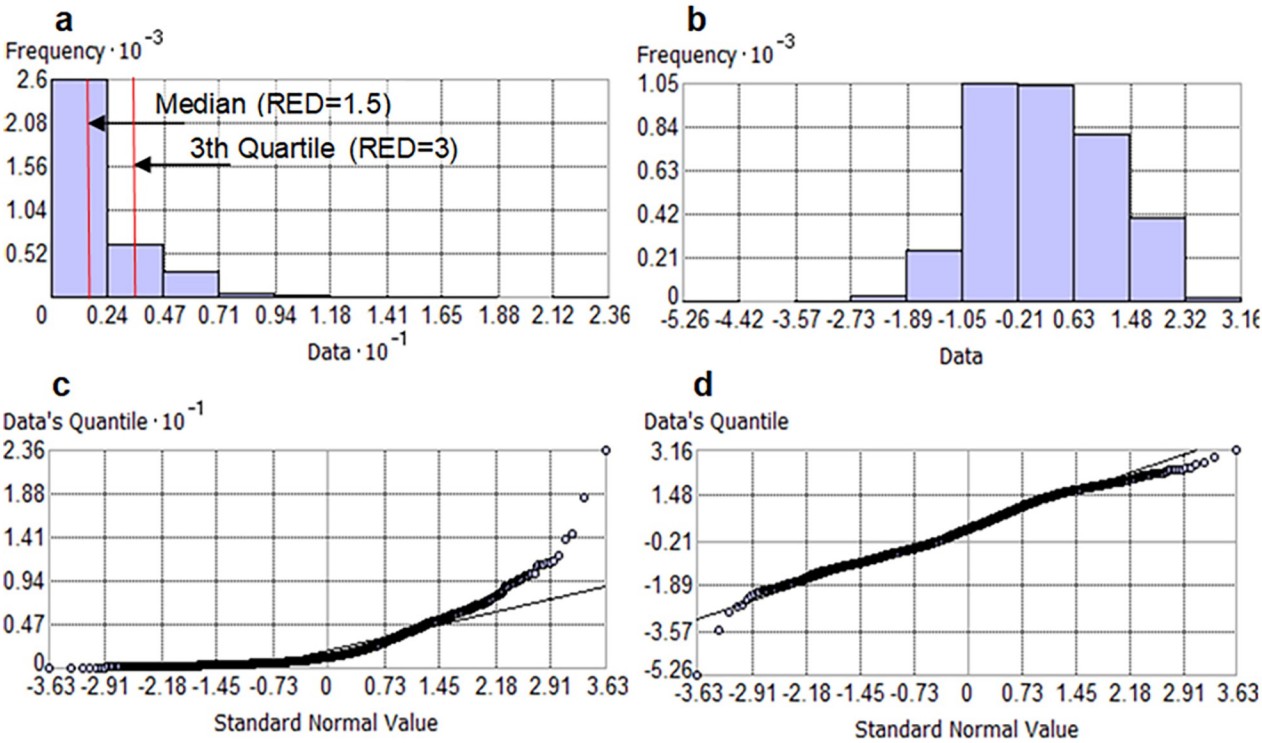

**Fig 4. Exploratory data analysis and verification of normality of rainfall erosivity density (RED) data.** a) Distributional frequency on the original RED data and b) after log-normal transformation; c) QQ-Plot of the theoretical and estimated distribution of the original RED data and d) after log-normal transformation. Warning and alert thresholds (MJ hm$^{-2}$ h$^{-1}$) are shown in a).

A lognormal transformation was used to approximate the skewed distribution of the RED data (**Fig 4A**), with $g = 2.244$ to a normal distribution (**Fig 4B**), with $g = 0.011$.

The QQ-plot shows the theoretical and estimated distribution deviating from the normal distribution as the RED values increase (**Fig 4C**), while a normalisation of the distribution is evident after the logarithmic transformation (**Fig 4D**). With the normality of the distribution restored, it is possible to extend the log-normal ordinary kriging to probability mapping [73].

The drift analysis demonstrated the existence of a non-random (deterministic) component in spatial distribution of the data: the moderate gradient of RED data occurs from north to south regions of the world. However, we considered that the stationarity hypothesis does not hold for the global, but only locally. Thus, in order to make a robust assumption of homogeneity of variances, the concept of process stationarity has been replaced by a stationarity of governing influence regarding local hydrological processes and nearby local anisotropy. In such a situation, ordinary kriging is recommended for interpolation [74].

## Spatial structural modelling

To instruct the kriging interpolation, a regionalisation model was fitted using an iterative procedure developed by Johnston et al. [73], which consists of two steps. Step 1 assumes an isotropic model, and performs an initial run of the experimental spatial structure on the standard deviation-scaled data $z(\mathbf{s}_0) = (z(\mathbf{s}_0) - \bar{z}) \cdot \sigma^{-1}$, where $z(\mathbf{s}_0)$ is used to denote the $j^{\text{th}}$ measurement of a variable at the $\alpha^{\text{th}}$ spatial locations $\mathbf{s}_0$, and $\sigma$ is the sample standard deviation. With step 2 any parameter is calibrated interactively, such as: *lag* number (assumed equal to 7), *lag* size **h** (assumed equal to 10 km for the warning state and 20 km for the alert state), *range* **a**

representing the limit of spatial dependence (equal to 112 km for the warning state and 70 km for the alert state, which are comparable to the ranges of spatial dependence of extreme precipitation on a global scale, varied between 54 and 265 km [75].

Isotropic semivariograms were then modelled as a combination of two distinct spatial structures, the nugget variance and a spherical structure, as shown in Eq (4):

$$\gamma(h) = \begin{cases} 0 & h = 0 \\ C_0 + C\left(\dfrac{3}{2}\dfrac{h}{a} - \dfrac{1}{2}\dfrac{h^3}{a^3}\right) & 0 < h \leq a \\ C_0 + C & h > a \end{cases} \tag{4}$$

The nugget effect $C_0$, equal to 0.066 for the warning state and 0.072 for the alert state, is simply the sum of the measurement error, and the microscale variation of RED, which remains unknown due to the spatial variability associated with the distance between raingauge stations. The value that the semivariogram model reaches at the *range* (the value on the h-axis) is called the *sill* (*partial sill + nugget*), and counts 0.182 for the warning state and 0.189 for the alert state, while $h$ is the distance between the unknown point ($h = 0$) and a generic point-station.

In this way, **Fig 5** shows the experimental unidirectional semivariogram computed from the 3,625 data of RED, with spherical admissible models fitted for threshold-values $z_k$ (RED > 1.5, and > 3.0 MJ hm$^{-2}$ h$^{-1}$).

Semivariogram values increase with separation distance, reflecting the assumption that RED data that are close tend to be more similar than data that are farther away. In particular, the spherical semivariogram model fluctuates around the *sill* value at 1.05° (~112 km) for the warning state (**Fig 5A**, violet curve) and at 0.70° (~70 km) for the alert state (**Fig 5B**, violet curve), suggesting that the phenomenon recorded at alert state operates on a smaller spatial scale than in the warning state. This is physically correct as RED events affect a smaller area as they become more extreme, and the physical mechanisms of extreme storm events are size-dependent [76].

The errors involved in the transfer of information from the point to the landscapes via LNOKpm were assessed by means of quantitative standard error estimation and cross-validation [77], re-estimating the RED data at raingauge locations after removing one RED value at a time from the datase [78]. The difference between the estimated and the corresponding actual indicator value is the experimental error. Thus, repeating this estimation for the number of the experimental data $n = 3,625$, the cross-validation statistics were calculated, as mean error and root mean square errors (RMSE).

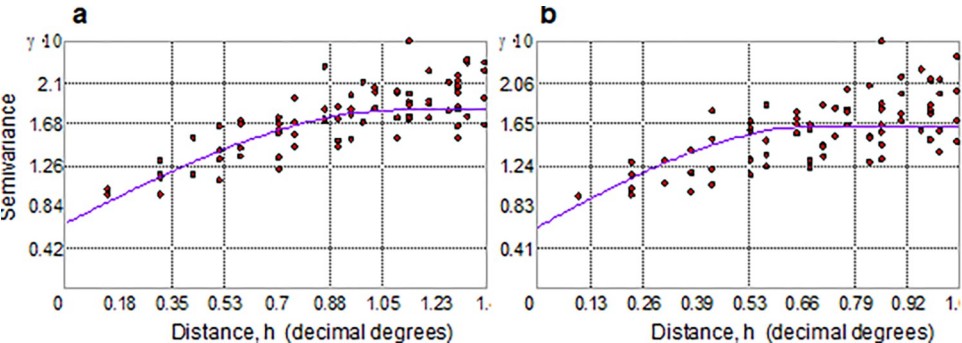

**Fig 5. Modelling of spatial dependence to instructing kriging interpolation.** Experimental semivariogram (dots) with permissible spherical model estimates (violet curve) at a) threshold $z_k > 1.5$ MJ hm$^{-2}$ h$^{-1}$ and b) at threshold $z_k > 3.0$ MJ hm$^{-2}$ h$^{-1}$ (**b**). Units of the semivariance $\gamma$ are multiplied by 10.

Cross-validation estimates the proportion of neighbourhood values that are below or above the threshold value. The high proportions of low values not exceeding the threshold and high values exceeding the threshold are a measure of the success of the kriging estimates Fig 6.

This is an estimate of the proportion of the values in the neighbourhood that are above the threshold value. The mean error values of -0.005 (warning state) and -0.019 (alert state), and RMSE = 0.262 (warning state) and RMSE = 0.215 (alert state), are close to zero, making it clear that there are no systematic errors.

## Global spatial pattern of RED-kriged probability estimation

Fig 7A presents the map of the hydrological hazard-prone areas at the warning state, highlighting the fact that about 31% of the world's land area has a greater than 50% probability of exceeding the $z_k$ threshold (RED > 1.5 MJ hm$^{-2}$ h$^{-1}$). The map indicates that the phenomenon taken into account by LNOKpm is not smooth (i.e. RED values change strongly with distance). The most affected hydrological hazard-prone areas are Africa and the southern Asian continents, southern Saudi Arabia, Australia, almost all of the USA with an offshoot to western Canada. These figures are consistent with statistics indicating that about 10 million hectares of cropland are lost each year due to soil erosion worldwide [79], particularly in Asia, Africa and South America, where erosion is more severe [80, 81].

China's far southeast, where the probability of exceeding RED at warning state is high, has experienced significant upward trends in rainfall erosivity over the period 1950–2010 [82]. Similarly, in many parts of Africa, where the warning state is expected to be exceeded over a wide area, soil erosion is becoming a major problem due to the high sediment production in tropical mountain streams [83]. The Mediterranean also has a high probability of reaching a warning state, where the aggressiveness of rainfall [8], in tandem with RED, seems to show a propensity to increase in recent decades [18]. However, in addition to the frequent high erosivity of precipitation regimes and human disturbances, it can be argued that much of the Mediterranean landscape is naturally vulnerable to soil erosion processes [84, 85]. Fig 7B, on the other hand, shows the map of the hydrological hazard-prone areas at alert state, highlighting the fact that ~19% of the world's land area has a greater than 50% chance of exceeding the $z_k$ threshold (RED > 3.0 MJ hm$^{-2}$ h$^{-1}$).

With respect to the alert threshold (Fig 7B), regions that have become free hazard-areas include the Mediterranean lands, almost all of the USA, Japan, Pakistan, northern and central

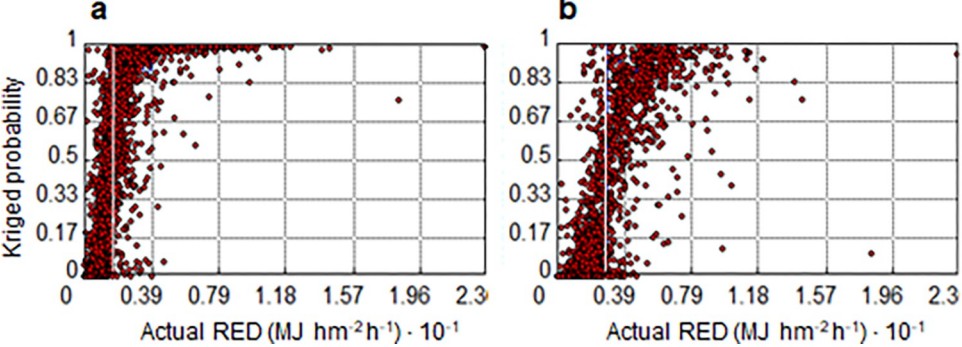

**Fig 6. Cross-validation for warning and alert states.** Scatterplots between actual rainfall erosivity density (RED) values above the given threshold and LNOKpm probability for the thresholds a) $z_k$ (RED) > 1.5 MJ hm$^{-2}$ h$^{-1}$ and b) $z_k$ (RED) > 3.0 MJ hm$^{-2}$ h$^{-1}$. The white vertical lines in both graphs represent the respective RED thresholds (a, warning state; b, alert state). The cross-validation scatter diagrams (**a and b**) show that the actual RED values below and above the given thresholds at the warning and alert states are in agreement with the respective kriged probability.

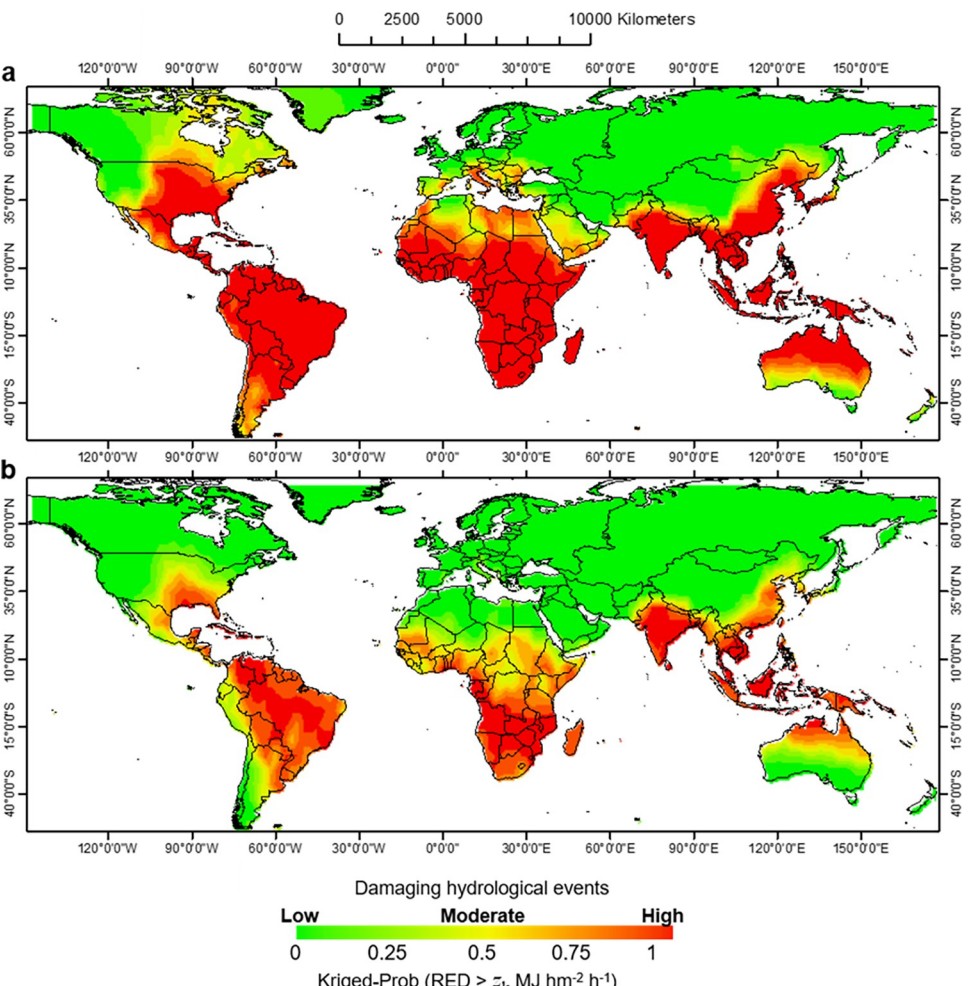

**Fig 7. Global spatial patterns of kriged-probability map over the period 2002–2011.** Exceedance of the rainfall erosivity density (RED) threshold-value at a) warning state: $z_k$ (RED) > 1.5 MJ hm$^{-2}$ h$^{-1}$, and b) alert state: $z_k$ (RED) > 3.0 MJ hm$^{-2}$ h$^{-1}$.

Africa and south-central Australia. Canada, Greenland and Eurasia still remain below the $z_k$ threshold in warning and alert states, because these regions have a climate less exposed to the critical level of RED, and where rainfall erosivity is also lower with a decrease in rainfall intensity as latitude increases northwards. For the areas of the USA exposed to hydrological hazards at the alert stage, the map in **Fig 7B** roughly overlaps with the map of the country highlighting the hurricane-prone areas [86].

The southernmost part of China, and parts of the southern USA, central and southern Africa, Latin America and India have a very high probability of exceeding the warning threshold. These regions reach soil erosion hotspots of > 20 Mg hm$^{-2}$ yr$^{-1}$, and are among that most intensely eroded areas in the world [87]. In particular, the observation in India follows the small convective systems that dominate throughout the Western Ghats region [88], but large events are also more intense [89].

These findings are consistent with the results of Medeiros et al. [90], who showed that the most perturbed areas in the tropical region are associated with convective storms that can have smaller radii (~10 km), as suggested by the smallest kriged-range obtained with the

semivariogram function exceeding the $z_k$ threshold (RED > 3.0 MJ hm$^{-2}$ h$^{-1}$) at alert state. These are the highest RED values found in most of South America and central and southern Africa, which are characterised by a complex property in the transfer of erosive energy to land. In these regions, sub-grid scale convection and intensification of rainfall generation processes are very hazardous, as long-lived mesoscale convective systems are well organised at these latitudes and contribute disproportionally to extreme tropical precipitation, with ~40% of days with more than 250 mm of rain over land being associated with convective systems lasting more than 24 hours [91].

## Kriged-probability mapping validation at continental spatial scale

The geostatistical approach used is robust to outlier effects even when the number of experimental data is relatively small and irregularly distributed [92], as the few network stations over some regions (e.g. Russia and the African continent). Then, a main drawback that can occur is when rainfall decreases significantly after heavy rainstorms [93]. However, our approach has proven to give satisfactory results when comparing LNOK-based probability maps with the hydrological disasters recorded by the Munich Reinsurance Company (Munich Re, https://www.munichre. com, on a continental scale (data not shown). Thus, for the Asian continent, it was possible to compare the kriged-probability map, for RED > 1.5 MJ hm$^{-2}$ h$^{-1}$, with the damaging hydrological events (storm + erosional soil degradation) that occurred between 1981 and 2018 (**Fig 8**).

The orange and red colours of the kriged map covering Saudi Arabia, Yemen, Oman, Pakistan, Nepal, Bangladesh, Burma, Vietnam, Thailand, Cambodia, the Indonesian Archipelago, southern Japan and south-eastern China (**Fig 8A**, red areas) are roughly overlapping with the hazards in multiple locations with the aggregate hydrological impact of the Munich Re dataset (not shown).

The average number of weather, climate and water hazards per decade has increased over the period 1970–2019 in Asia (**Fig 8B**). They have increased from, on average, one disaster every 15 days to one every three days [94], with a higher proportion of floods and storm events (**Fig 8C**). On the central and northern Asian continent, the low probability of high RED values (**Fig 7**), generally associated with a low probability of damaging hydrological events (**Fig 8A**), is consistent with the rainfall minima of desert areas (e.g. in Mongolia and in northwestern China) and Siberia [95].

For the African continent, spatial dependence may be more difficult to detect because RED data are scarce. Although broad climatic patterns can be identified across the African continent, there are many local variations from place to place, with the most important differentiating climatic factor being rainfall [96]. The continent is most affected by both a continental tropical air mass to the north and by maritime tropical and equatorial air masses to the south (meeting in the Inter-Tropical Convergence Zone) [97]. Essentially, the equatorial maritime air mass is unstable and brings rain, while the tropical maritime air mass, when fully developed, is stable and generally does not bring rain unless forced over a high mountain [98].

However, with the exception of some northern coastal countries and an overestimation for some southern countries, it is still possible to roughly identify the areas most susceptible to damaging hydrological events for African lands below 15˚ N (**Fig 8D**, red areas), compared to the flood pattern of the Munich Re dataset (not shown).

This is also in line with increased flooding and storm-driven erosion in Africa (**Fig 8E**), and nutrient loss due to increased extreme diurnal rainfall observed in tropical eastern and southeastern Africa in the late 20[th] century [99], calling into question the sustainability of food security for the ~300 million people currently living in Africa south of the equator [100]. Floods account for 60% of recorded disasters, while storms account for 17% (**Fig 8F**).

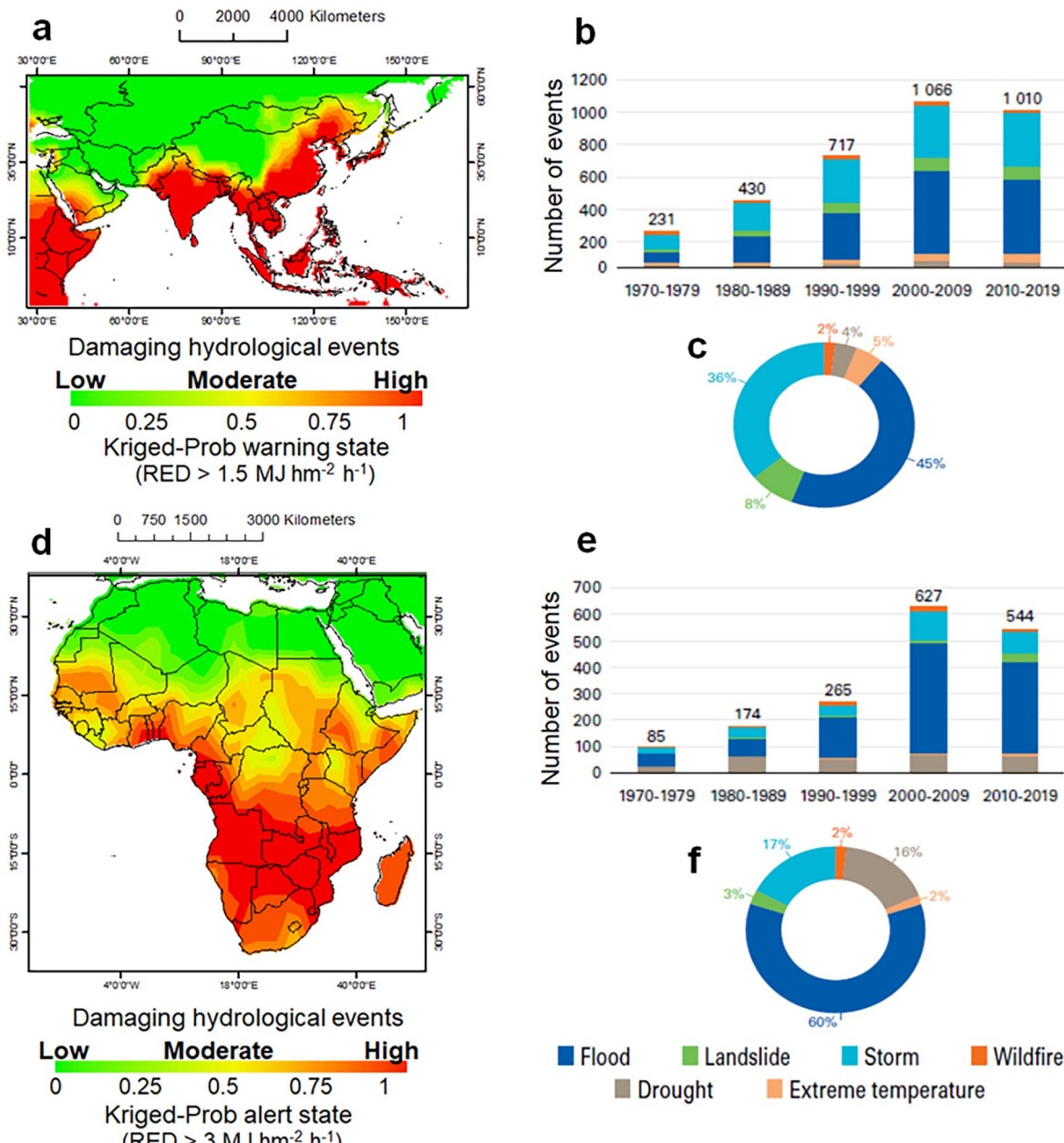

**Fig 8. Comparison of kriged probability maps with damaging hydrological events (Asia) and flood events (Africa) over the last four-five decades.** a) Rainfall erosivity density (RED) > 1.5 MJ hm$^{-2}$ h$^{-1}$ with b) with trend of disasters and c) related percentage; d) Rainfall erosivity density (RED) > 3.0 MJ hm$^{-2}$ h$^{-1}$ with e) trend of disasters and f) related percentage. The graphs a) and d) refer to the period 2002–2011, graphs b), c), e) and f) to 1971–2019.

## Conclusion

Detailed rainfall datasets and appropriate modelling approaches are necessary to establish the mechanisms underlying the complex set of physical processes that govern the response of hydrological cycles to a changing climate. Indeed, hydrological hazard responses to climate change remain difficult to quantify with existing modelling frameworks. Although in some areas of the world (such as the African continent) the network of weather stations is still

insufficient, with the widespread availability of high-temporal resolution rainfall records for large areas and modern advances in climate modelling, new opportunities are opening up for the use of geostatistical methods for large-scale planning, hazard management and risk prevention. This paper presents a geostatistical modelling framework for the proper interpretation of spatially-explicit hydrological hazard, which assumes a set of quantitative data for the location of interest (precipitation extremes and rainfall erosivity) and probabilities associated with ranges of rainfall erosivity density above critical values. They include the best available precipitation data from a set of stations (3,625) worldwide (63 countries) and rainfall erosivity data, as provided by an updated version of the Global Rainfall Erosivity Database.

In this study, we have analysed for the first time the spatial pattern of hydrological hazard associated with rainfall erosivity in a global-scale visualisation. The results indicated that about 31% and 19% of the world's land area have a greater than 50% probability of exceeding the warning and alert thresholds of 1.5 and 3.0 MJ hm$^{-2}$ h$^{-1}$, respectively, with the most affected regions being tropical Latin America, South Africa, India and the Indian Archipelago. The geostatistical modelling, designed for a spatial resolution of ~100 km, is compatible with the vast majority of countries in the world (167 out of 234 having an area >10,000 km$^2$, https://www.worldometers.info/geography/largest-countries-in-the-world) and within the aggregation range of most environmental and biodiversity models [101]. Highlighting the potential of probabilistic geostatistical modelling, our results suggest the possibility of using geostatistical spatial modelling to determine the probability of exceeding thresholds of high erosivity density and to generate probability maps to delineate the most sensitive areas, which may lead to catastrophic regime shifts related to the occurrence of damaging hydrological events. This soft-computing modelling represents a paradigm shift on how to provide timely, accurate and actionable information on hydrological hazards [102–104]. Without giving the value at each point but returning a probability map, this approach offers the possibility to obtain information also where no data or measurements exist, as it identifies the hydrological hazard associated with the probability of exceeding an erosivity density threshold. In this way, the approach can support decision-making. We thus offer these results as a springboard to support policymakers, local authorities and civil protection in planning medium- and long-term actions to reduce hydrological disasters [105–106]. As rainfall erosivity is projected to increase by at least 35% globally by 2070 [107], the probability of hydrological hazards will also show similar trends.

A more careful validation of the global map of hydrological disaster-prone areas is certainly needed. However, this is a promising first step, and the global probability map was well suited to hydrological disasters in regions where data coverage was substantial. Quantifying the probability of exceeding threshold values of erosivity density in a way that enables meaningful comparisons with hydrological records is an important topic of study, and our article is a step forward towards this goal. In fact, geostatistical methods can be practically implemented to create spatially explicit probabilistic maps at the country level, and can be useful in the study of erosive hazards, which, however, depend on several interacting factors, such as complex orography, large-scale air flow and teleconnection patterns. Then, purely geostatistical findings do not produce an explicit mechanistic modelling of rainfall erosivity density, which, however, is data demanding and can be accompanied by a large amount of uncertainties in the estimates. Future studies should assess the results in diverse physical geographic conditions and socio-economic situations, taking into account that population density, infrastructures, plant density and other factors also influence the occurrence of damage. In addition, probability calculations have also to take into account particularly long periods of low rainfall intensity, which are not erosive but can lead to deadly flooding and landslides.

## Acknowledgments

The authors would like to acknowledge the following services for proving their data: Bureau of Meteorology in Australia, New Zealand Institute of Water and Atmospheric Research (NIWA), Japan Meteorological Agency (JMA), Korea Meteorological Administration (KMA), National Meteorological Information Center (NMIC)of China, India Meteorological Department (IMD) of the Ministry of Earth Sciences (MoES), Iranian Meteorological Organization (IMO), Directorate General of Civil Aviation (DGCA)—State of Kuwait, Lomonosov Moscow State University (LMSU), Israel Meteorological Service (IMS), Turkish Ministry of Forestry and Water Affairs (MoFWA), U.S. Climate Reference Network (USCRN) and National Oceanic and Atmospheric Administration (NOAA), Comisión Nacional Del Agua (CONAGUA) —Servicio Meteorologico Nacional (SMN) of Mexico, Meteorological Service of Jamaica (MSJ), University of Costa Rica (UCR), Centro Nacional de Investigaciones de Café (CENI-CAFÉ) of Colombia, General Directorate of the Water Resources (GDWR) of Chile, Meteorological Department Suriname (MDS), Mauritius Meteorological Services (MMS), Algerian National Agency for Hydraulic Resources (ANRH).

## Author Contributions

**Conceptualization:** Gianni Bellocchi.

**Data curation:** Nazzareno Diodato, Gianni Bellocchi.

**Formal analysis:** Nazzareno Diodato, Gianni Bellocchi.

**Funding acquisition:** Panos Panagos.

**Investigation:** Nazzareno Diodato.

**Methodology:** Pasquale Borrelli, Panos Panagos.

**Project administration:** Panos Panagos.

**Resources:** Nazzareno Diodato, Pasquale Borrelli.

**Software:** Nazzareno Diodato.

**Supervision:** Pasquale Borrelli, Panos Panagos, Gianni Bellocchi.

**Validation:** Nazzareno Diodato, Pasquale Borrelli.

**Visualization:** Nazzareno Diodato.

**Writing – original draft:** Nazzareno Diodato, Pasquale Borrelli, Panos Panagos, Gianni Bellocchi.

**Writing – review & editing:** Nazzareno Diodato, Pasquale Borrelli, Panos Panagos, Gianni Bellocchi.

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
