## [Decision Letter · Decision Letter 0]

4 Apr 2022

PONE-D-22-06568Global assessment of hydrological disaster prone-areasPLOS ONE

Dear Dr. Panagos,

Thank you for submitting your manuscript to PLOS ONE. After careful consideration, we feel that it has merit but does not fully meet PLOS ONE’s publication criteria as it currently stands. Therefore, we invite you to submit a revised version of the manuscript that addresses the points raised during the review process.

We look forward to receiving your revised manuscript.

Kind regards,

Chun Liu

Academic Editor

PLOS ONE

Journal Requirements:

"NO funding was received. The Open access will be paid by the Corresponding author institution"

3. We note that Figures 3, 4 and 5 in your submission contain [map/satellite] images which may be copyrighted. All PLOS content is published under the Creative Commons Attribution License (CC BY 4.0), which means that the manuscript, images, and Supporting Information files will be freely available online, and any third party is permitted to access, download, copy, distribute, and use these materials in any way, even commercially, with proper attribution. For these reasons, we cannot publish previously copyrighted maps or satellite images created using proprietary data, such as Google software (Google Maps, Street View, and Earth). For more information, see our copyright guidelines: http://journals.plos.org/plosone/s/licenses-and-copyright.

a. You may seek permission from the original copyright holder of Figures 3, 4 and 5 to publish the content specifically under the CC BY 4.0 license.  

Reviewers' comments:

Reviewer's Responses to Questions

**Comments to the Author**

1. Is the manuscript technically sound, and do the data support the conclusions?

Reviewer #1: Yes

Reviewer #2: Yes

Reviewer #3: Yes

2. Has the statistical analysis been performed appropriately and rigorously? 

Reviewer #1: Yes

Reviewer #2: Yes

Reviewer #3: Yes

3. Have the authors made all data underlying the findings in their manuscript fully available?

Reviewer #1: Yes

Reviewer #2: Yes

Reviewer #3: Yes

4. Is the manuscript presented in an intelligible fashion and written in standard English?

Reviewer #1: Yes

Reviewer #2: Yes

Reviewer #3: Yes

5. Review Comments to the Author

Reviewer #1: Dear authors

The Manuscript is valuable and can be considered by different researchers of the world.

They have investigated Rainfall erosivity density (RED), i.e. rainfall erosivity (MJ mm hm -2 h -1 yr -1 ) per rainfall unit (mm) in worldwide using log-normal ordinary kriging with probability mapping. Then they identified damaging hydrological hazard prone areas that exceed warning and alert thresholds.

congratulations.

Reviewer #2: This is a well-written paper discussing the global outlook of the erosivity density (RED), as an indicator of hydrology-related disaster-prone areas. I provide some specific comments below for the authors to address:

1) I believe the current title is too broad to be covered by the RED-based analysis. The term "hydrological disaster" is too general in my opinion; not only the quantity, but also the quality point of view can be assessed. Even if we focus on only quantity, not only too much water (e.g., flood) but also too little water (e.g., drought) can lead to a disaster. The authors should revise the title to better reflect what they really present and discuss in the present study. By the way, it should be disaster-prone areas instead of disaster prone-areas.

2) Similar to the issue in the title, we can also find some descriptions that are not so accurate in the text. For instance, Line 31, I don't think RED is a straightforward indicator of storm surges as surges depend (more) on other factors such as sea level and tidal activity. The authors should correct this kind of inaccuracy throughout the text.

3) I don't mean to be picky, but based on the main result in Fig. 3, I am not in full agreement with the referred "global" assessment. Apparently there is a big chunk of regions left out because of the short in ground observations (and/or other reasons). In fact, since Africa is blacked out, I don't understand how you could state something like Line 125 that the most affected hydrological hazard-prone areas are Africa, etc.

4) Following my point above, I don't feel right to select Africa for validation, as shown in Fig. 4. There is nothing we can compare. North or South America and Australia should be a better region for the task like this.

5) Another point related to 3), why is the (nearly) entire European territory left out in Fig. 3, but apparently data is not an issue as shown in Fig. 5? A minor point here regarding Fig. 5 is the weird "arc-like" symbol while the caption tells "circles."

6) The authors should discuss how changes in the coefficient in the RUSLE equation affect their assessment. A similar question can be applied to the deterministic RED thresholds (i.e., 1.5 and 3). Some sensitivity analysis is required.

7) What is the spatial resolution of the RED-based assessment? Can the assessment provide fine enough details for regions at a relatively smaller spatial scale (e.g., a country)? The aspect of usefulness (e.g., pertaining to Fig. 3) should also be addressed.

Reviewer #3: Rainfall erosivity is a critical factor that affect human societies and natural landscapes. Global Rainfall Erosivity Database was used, this paper tried to assess the global hydrological disaster prone-areas. The work is important and has its value to enhance our understanding on the spatial characteristics of soil erosion and disasters.

I suggest the authors can further select some references which support these conclusions, Since some of them may go beyond the research content, et al. the explanations of regional intense RED, making the conclusions of this paper weaker. And there was a little mess about the results and discussion. I would recommend clarifying sections of the results and “discussion and look a forward” before publishing this manuscript. It needs major revision before it could be considered for publication.

7. Line 281. Had the data been downscaling?

6. PLOS authors have the option to publish the peer review history of their article (what does this mean?). If published, this will include your full peer review and any attached files.

Reviewer #1: No

Reviewer #2: No

Reviewer #3: No

---

## [Author Response · Author response to Decision Letter 0]

13 May 2022

Panos Panagos 

Journal Requirements:

Our Reply: The resubmitted manuscript conforms to the journal’s style requirements.

"NO funding was received. The Open access will be paid by the Corresponding author institution"

Our Reply: Please consider the following statements by the authors:

"NO funding was received. The Open access will be paid by the Corresponding author institution".

“The authors received no specific funding for this work”.

3. We note that Figures 3, 4 and 5 in your submission contain [map/satellite] images which may be copyrighted. All PLOS content is published under the Creative Commons Attribution License (CC BY 4.0), which means that the manuscript, images, and Supporting Information files will be freely available online, and any third party is permitted to access, download, copy, distribute, and use these materials in any way, even commercially, with proper attribution. For these reasons, we cannot publish previously copyrighted maps or satellite images created using proprietary data, such as Google software (Google Maps, Street View, and Earth). For more information, see our copyright guidelines: http://journals.plos.org/plosone/s/licenses-and-copyright.

Our Reply: We have removed Munich Re’s data and maps (former Fig. 1a, Fig. 4a, b) because they are not available for the time being and the company cannot provide permission for their publication. This implies that we refer to Munich Re’s data as “not shown” (line 337, line 347).

 

Reviewer #1: Dear authors

The Manuscript is valuable and can be considered by different researchers of the world.

They have investigated Rainfall erosivity density (RED), i.e. rainfall erosivity (MJ mm hm -2 h -1 yr -1 ) per rainfall unit (mm) in worldwide using log-normal ordinary kriging with probability mapping. Then they identified damaging hydrological hazard prone areas that exceed warning and alert thresholds.

congratulations.

Our Reply: We are grateful for recognising the value of our submission.

 

Reviewer #2: This is a well-written paper discussing the global outlook of the erosivity density (RED), as an indicator of hydrology-related disaster-prone areas. I provide some specific comments below for the authors to address:

Our Reply: We are grateful for recognising the importance of our submission and for providing comments to improve it.

1) I believe the current title is too broad to be covered by the RED-based analysis. The term "hydrological disaster" is too general in my opinion; not only the quantity, but also the quality point of view can be assessed. Even if we focus on only quantity, not only too much water (e.g., flood) but also too little water (e.g., drought) can lead to a disaster. The authors should revise the title to better reflect what they really present and discuss in the present study. By the way, it should be disaster-prone areas instead of disaster prone-areas.

Our Reply: New title: “Global assessment of storm disaster-prone areas”.

2) Similar to the issue in the title, we can also find some descriptions that are not so accurate in the text. For instance, Line 31, I don't think RED is a straightforward indicator of storm surges as surges depend (more) on other factors such as sea level and tidal activity. The authors should correct this kind of inaccuracy throughout the text.

Our Reply: This is right because "storm surges" is not part of our storm disaster series. Inadvertently reported in the original submission, it was deleted in the revised manuscript.

3) I don't mean to be picky, but based on the main result in Fig. 3, I am not in full agreement with the referred "global" assessment. Apparently there is a big chunk of regions left out because of the short in ground observations (and/or other reasons). In fact, since Africa is blacked out, I don't understand how you could state something like Line 125 that the most affected hydrological hazard-prone areas are Africa, etc.

Our Reply: We can state that “Africa and the southern Asian continents, southern Saudi Arabia, Australia, almost all of the USA with an offshoot to western Canada” (lines 286-288) are the areas most affected by storm hazards, as the red colours in the map of Fig. 7a indicate high values of the kriged probability. Backed out Africa (but also parts of Europe and South America) may be due to the loss of colour when converting the .docx file to .pdf during the initial submission process.

4) Following my point above, I don't feel right to select Africa for validation, as shown in Fig. 4. There is nothing we can compare. North or South America and Australia should be a better region for the task like this.

Our Reply: The answer to this remark is linked to the previous one.

5) Another point related to 3), why is the (nearly) entire European territory left out in Fig. 3, but apparently data is not an issue as shown in Fig. 5? A minor point here regarding Fig. 5 is the weird "arc-like" symbol while the caption tells "circles."

Our Reply: The answer to this remark is linked to the previous one.

6) The authors should discuss how changes in the coefficient in the RUSLE equation affect their assessment. A similar question can be applied to the deterministic RED thresholds (i.e., 1.5 and 3). Some sensitivity analysis is required.

Our Reply: Since the aim of our study is to assess storm hazards from an actual dataset on rainfall and rainfall erosivity around the world, an assessment on the impact of changes in the erosivity factor seems out of place here. In fact, rainfall erosivity is known to be the most important climatic factor in the (R)USLE approach, and how its changes impact on soil erosion and other damaging storm events such as floods and flash-floods is not discussed again. Then, a sensitivity analysis on thresholds would involve checking the probability of exceeding each threshold on a gradient of thresholds. However, the question does not arise here because ours are specific thresholds that we have identified among the n possible ones that are able to highlight the probability of occurrence of given hydrogeological events (e.g. floods, soil erosion). Testing geostatistical modelling with other threshold values becomes a redundant and superfluous task (as a geostatistical model is required for each threshold), as our thresholds are statistically relevant as they “correspond, respectively, to the median and the 3rd quartile of the distribution of the RED data (Fig. 4a)” (lines 169-170) and “also mark the range of critical RED values identified by Diodato et al. [8] with a 50-year return period” (lines 170-171). Representing probability maps with values other than the selected thresholds would not add any relevant information to this study, apart from not revealing the purpose for which the critical thresholds were set.

7) What is the spatial resolution of the RED-based assessment? Can the assessment provide fine enough details for regions at a relatively smaller spatial scale (e.g., a country)? The aspect of usefulness (e.g., pertaining to Fig. 3) should also be addressed.

Our Reply: “The geostatistical modelling, designed for a spatial resolution of ~100 km, is compatible with the vast majority of countries in the world (167 out of 234 having an area >10000 km2, https://www.worldometers.info/geography/largest-countries-in-the-world) and within the aggregation range of most environmental and biodiversity models [101]” (lines 394-397). This can be seen in the semivariograms in Fig. 5a, b, where in fact the semivariance is constant just after a distance (h) of “1.05° (~112 km) for the warning state … and at 0.70° (~70 km) for the alert state” (lines 253-254). Based on the pattern of available stations, it was possible to go down to this spatial detail although the model works on a global scale. This downscaling has a positive effect on the assessment of storm disaster-prone areas around the world and helps us to better understand the geographical aspects of flooding, soil erosion and landscape transformation (e.g. Amatulli et al., 2018, Scientific Data, https://doi.org/10.1038/sdata.2018.40).

 

Reviewer #3: Rainfall erosivity is a critical factor that affect human s!ocieties and natural landscapes. Global Rainfall Erosivity Database was used, this paper tried to assess the global hydrological disaster prone-areas. The work is important and has its value to enhance our understanding on the spatial characteristics of soil erosion and disasters.

Our Reply: We are grateful for recognising the importance of our submission and for providing comments to improve it.

I suggest the authors can further select some references which support these conclusions, Since some of them may go beyond the research content, et al. the explanations of regional intense RED, making the conclusions of this paper weaker.

Our Reply: The added references 102-106 support our concluding statements. The following supported concluding statement was added: “The geostatistical modelling, designed for a spatial resolution of ~100 km, is compatible with the vast majority of countries in the world (167 out of 234 having an area >10000 km2, https://www.worldometers.info/geography/largest-countries-in-the-world) and within the aggregation range of most environmental and biodiversity models [101]”.

And there was a little mess about the results and discussion. I would recommend clarifying sections of the results and “discussion and look a forward” before publishing this manuscript. It needs major revision before it could be considered for publication.

Our Reply: The manuscript was restructured to conform to the style of the journal. The revised version includes a “Results and Discussion” section and a “Conclusion” section.

7. Line 281. Had the data been downscaling?

Our Reply: This is a downscaling study. Our text informs that we used “downscaled RED data from 3,615 raingauges worldwide and log-normal ordinary kriging with probability mapping” (lines 22-23). We added the following: “The geostatistical modelling, designed for a spatial resolution of ~100 km, is compatible with the vast majority of countries in the world (167 out of 234 having an area >10000 km2, https://www.worldometers.info/geography/largest-countries-in-the-world) and within the aggregation range of most environmental and biodiversity models [101]” (lines 394-397).

---

## [Decision Letter · Decision Letter 1]

14 Jul 2022

Global assessment of hydrological disaster prone-areas

PONE-D-22-06568R1

Dear Dr. Panagos,

We’re pleased to inform you that your manuscript has been judged scientifically suitable for publication and will be formally accepted for publication once it meets all outstanding technical requirements.

Kind regards,

Chun Liu

Academic Editor

PLOS ONE

Additional Editor Comments (optional):

Reviewers' comments:

Reviewer's Responses to Questions

**Comments to the Author**

1. If the authors have adequately addressed your comments raised in a previous round of review and you feel that this manuscript is now acceptable for publication, you may indicate that here to bypass the “Comments to the Author” section, enter your conflict of interest statement in the “Confidential to Editor” section, and submit your "Accept" recommendation.

Reviewer #1: All comments have been addressed

Reviewer #2: All comments have been addressed

2. Is the manuscript technically sound, and do the data support the conclusions?

Reviewer #1: Yes

Reviewer #2: Yes

3. Has the statistical analysis been performed appropriately and rigorously? 

Reviewer #1: Yes

Reviewer #2: Yes

4. Have the authors made all data underlying the findings in their manuscript fully available?

Reviewer #1: Yes

Reviewer #2: Yes

5. Is the manuscript presented in an intelligible fashion and written in standard English?

Reviewer #1: Yes

Reviewer #2: Yes

6. Review Comments to the Author

Reviewer #1: (No Response)

Reviewer #2: Thanks the authors for addressing my comments. I think this article is a great piece of work for publication.

7. PLOS authors have the option to publish the peer review history of their article (what does this mean?). If published, this will include your full peer review and any attached files.

Reviewer #1: No

Reviewer #2: No

---

## [Editor Report · Acceptance letter]

28 Jul 2022

PONE-D-22-06568R1 

Global assessment of storm disaster-prone areas 

Dear Dr. Panagos:

I'm pleased to inform you that your manuscript has been deemed suitable for publication in PLOS ONE. Congratulations! Your manuscript is now with our production department. 

Kind regards, 

on behalf of

Dr. Chun Liu 

Academic Editor

PLOS ONE